# Autophagy Pathways in the Genesis of *Plasmodium*-Derived Microvesicles: A Double-Edged Sword?

**DOI:** 10.3390/life12030415

**Published:** 2022-03-12

**Authors:** Inès Leleu, Jeremy Alloo, Pierre-André Cazenave, Jacques Roland, Sylviane Pied

**Affiliations:** U1019-UMR 9017-CIIL-Center for Infection and Immunity of Lille, Institut Pasteur de Lille, CHU Lille, Inserm, CNRS, Université Lille, F-59000 Lille, France; ines.leleu@inserm.fr (I.L.); jeremy.alloo@pasteur-lille.fr (J.A.); cazenave@pasteur.fr (P.-A.C.); jacques.roland@pasteur-lille.fr (J.R.)

**Keywords:** astrocytes, autophagy, microvesicles, *Plasmodium*, cerebral malaria, neuroinflammation, pathophysiology

## Abstract

Malaria, caused by *Plasmodium* species (spp.), is a deadly parasitic disease that results in approximately 400,000 deaths per year globally. Autophagy pathways play a fundamental role in the developmental stages of the parasite within the mammalian host. They are also involved in the production of *Plasmodium*-derived extracellular vesicles (EVs), which play an important role in the infection process, either by providing nutrients for parasite growth or by contributing to the immunopathophysiology of the disease. For example, during the hepatic stage, *Plasmodium*-derived EVs contribute to parasite virulence by modulating the host immune response. EVs help in evading the different autophagy mechanisms deployed by the host for parasite clearance. During cerebral malaria, on the other hand, parasite-derived EVs promote an astrocyte-mediated inflammatory response, through the induction of a non-conventional host autophagy pathway. In this review, we will discuss the cross-talk between *Plasmodium*-derived microvesicles and autophagy, and how it influences the outcome of infection.

## 1. Introduction

Malaria, a parasitic disease caused by *Plasmodium* spp., resulted in more than 229 million clinical cases globally in 2019 [1]. The infection starts with a bite from an infected female *Anopheles* mosquito, which injects thousands of sporozoites into the mammalian host derma. The sporozoites rapidly travel to the liver, via the bloodstream, to initiate the pre-erythrocytic stage of the infection. They infect the hepatocytes, transform into trophozoites, and, eventually, give rise to thousands of merozoites. The released merozoites invade red blood cells (RBCs) to initiate the erythrocytic stage of the infection, responsible for clinical manifestations of malaria [2,3]. The adhesion and sequestration of the infected RBCs (iRBCs) in various organs is responsible for the severe manifestations of the disease [4]. From the asymptomatic liver stage to the clinical erythrocytic stage, intracellular *Plasmodium* parasites are always enclosed in the parasitophorous vacuole (PV). This PV is essential for growth and nourishment of the parasite. PV is also important in evading the host immune system, as well as escaping clearance by host autophagy mechanisms [5,6].

Autophagy is an intracellular vesicular-related process that regulates the cell environment against pathological conditions [7,8]. It consists of a highly conserved and synchronized network of autophagy-related (ATG) genes and proteins that promotes the formation of the microtubule-associated protein 1 light chain 3 (MAP1LC3/LC3)-positive phagophore, which becomes a double-membrane structured autophagosome through the acquisition of lipids [8,9,10]. Autophagy actively participates in cellular homeostasis, through the degradation, clearance, and recycling of damaged proteins and organelles from the cytoplasm to autophagosomes, and then to lysosomes [11,12,13]. 

The parasites release a large number of extracellular vesicles (EVs) during infection, and these contribute to inflammatory processes associated with malaria [14]. EVs are small, lipid bilayer membrane-bound vesicles that are naturally released from almost all types of cells, and they contain proteins, lipids, nucleic acids, and other metabolites [14,15]. They are formed under natural or pathological states during the process of endocytosis and/or autophagy [7,8,16]. They are classified into several subtypes, based on their size, mode of release, origin, and composition. Exosomes (40 to 120 nm in diameter) are the smallest vesicles, generated from endosome invagination [14,17,18]. Microvesicles (50 to 1000 nm in diameter) are formed by the budding of the plasma membrane [14,17,18]. Apoptotic bodies (2000 to 5000 nm in diameter) are the largest vesicles, derived from cells undergoing apoptosis [14,17]. EVs are involved in a wide range of biological processes, including cell-to-cell communication and transport, and the exchange of genetic information, cytosolic proteins, or lipids [18]. 

EVs have emerged as key players in most parasitic diseases, including malaria. They play an important role in infection biology and immunopathogenesis [14,15,19]. They are implicated in the infectious process, as they are involved not only in providing nutrients for parasite differentiation and multiplication, but are also important players in host–parasite interactions and the activation of the immune response [6,20]. In malaria, EVs are often referred to as microparticles (MPs), and can be of host or parasite origin [14]. Both infected and uninfected endothelial cells, reticulocytes, and platelets release MPs upon interaction with parasites [14,21,22]. EVs were first described in 2004 in Malawian children with cerebral malaria (CM), and, since then, several studies have described not only the presence of EVs during disease, but have also suggested their potential involvement in disease severity. EVs are, therefore, suggested to be a potential biomarker for the severity of the disease [14,23,24]. *Plasmodium*-derived microvesicles (pMVs) are thought to promote parasite virulence and influence pathogenesis, by modulating host immune responses through intercellular communication [25]. However, both the nature of the cargo and the mechanisms of actions of pMVs remain undefined. 

In this paper, we will discuss the cross-talk between pMVs and autophagy, and how this influences the outcome of the infection. We will first focus on the role of autophagy and pMVs in parasite growth and multiplication, before discussing the importance of autophagy in the transfer of pMVs to astrocytes and their role in the pathogenesis of CM.

## 2. Materials and Methods

### 2.1. Mice and Parasites

We used female C57BL/6 mice, 8–10 weeks old (Janvier laboratories, C57BL/6JRjFEMELLESPF8). C57BL/6 TLR3-deficient mice lines [26] and C57BL/6.WLA-Berr2 congenic mice ((B6.WLA-Berr1) ECM resistant CM^R^) [27] were bred and maintained under specific pathogen-free (SPF) conditions at Institute Pasteur Lille animal facility. Experiments were performed in agreement with the ethics of animal experimentation, and were approved by the French animal welfare committee “Ministère de l’Agriculture et de la Pêche” n°A 75485. All experiments carried out on CM^R^ and TLR3KO mice were conducted as described in the previous work published in CM^S^ [28]. Primary cultures of astrocytes and GFP-*Pb*A (1.49L clone, gift of Dr D. Walliker, Institute of Genetics, Edinburgh, UK) were used as previously described in [28].

### 2.2. Confocal and Transmission Electron Microscopy

Purified astrocytes from CM^R^ and TLR3KO mice were stimulated with GFP-*Pb*A-iRBCs (10:1) at 37 °C for 6 h and removed, then further incubated for 24 h or 48 h. For confocal microscopy, cells were plated on glass slides, and for TEM, on coverslips in 35 mm glass bottom dishes (No 1.5, MatTek, P35G-1.5-10-C), as previously described [28].

### 2.3. Quantification of Gene Expression

Gene expression quantification was performed using RT-qPCR RT-qPCR on 1 × 10^6^ cells. Cell treatment and processing were conducted as described by Leleu et al. [28].

## 3. Results

### 3.1. Host Autophagy Pathways in Malaria

The following two types of autophagy pathways are known to be engaged during the development of *Plasmodium* in the mammalian host: (a) Non-selective autophagy, which is an adaptive response for cellular remodeling upon unfavorable stress conditions. It is characterized by a double-membrane-bound autophagosome, decorated by microtubule-associated protein 1 light chain 3 beta (LC3-II) molecules. These autophagosomes engulf the parasite, and also support its nourishment during the intrahepatic stage [12,29]. (b) selective autophagy, which is initiated by unc-51-like autophagy-activating kinase-1 (Ulk) and class III phosphatidylinositol 3-kinase (PI3K) complexes. Selective autophagy specifically targets intracellular parasites and promotes their elimination through ubiquitination [8,30,31]. 

Recently, we have demonstrated the involvement of an unconventional LC3-mediated autophagy pathway (LAP), independent of Ulk1, in the transfer and degradation of *P. berghei ANKA* (*Pb*A)-MVs inside astrocytes. LAP is an intermediary pathway between autophagy and phagocytosis, involving ATG proteins and the recruitment of LC3-II directly on the single-membrane LAPosome formed around the engulfed microbe, prior to its fusion with lysosomes [32]. The LAP pathway was also found to be an inducer of astrocyte pro-inflammatory responses that play a major role in the pathogenesis of CM [28]. 

### 3.2. pMVs and Autophagy-Related Responses during Pre-Erythrocytic Development

During the liver stage of development, packed inside a transient vacuole (TV), the sporozoites injure and traverse several hepatocytes, before finally settling in one to initiate infection [2]. The invading sporozoite is contained in the PV, wherein it first differentiates into a trophozoite, before switching on replicative processes to form a schizont, containing thousands of merozoites. These merozoites egress the hepatocyte in a merosome (see below), invade erythrocytes, and start the blood-stage infection [2,33,34]. The two vacuolar organelles, TV and PV, constitute the first *Plasmodium*-derived vesicles during the parasitic life cycle. They allow the parasite (i) to escape hepatocyte elimination strategies, and (ii) to use host cell nutrients and develop alternative scavenging pathways for its survival [5,6,35,36]. As summarized in Figure 1A, the following two host cell pathways for clearing the parasite are triggered during *Pb*A sporozoite development in the liver: (1) in the early intrahepatic stage, parasites inside a TV, or in a deficient PV, are cleared by the induction of a PI3P-associated sporozoite elimination (PASE) process that differs from autophagy, but induces a lysosomal acidification pathway; (2) in the late intrahepatic stage, the PV membrane (PVM) engulfed parasite is targeted by LAP-like non-conventional autophagy that occurs independently of PI3K, RB1-inducible coiled-coil 1 (Rb1cc1), and Ulk1, and without the formation of reactive oxygen species (ROS) (Figure 1A) [37,38,39,40]. The absence of PI3K and Becn1 (beclin 1, autophagy related) in this process argues in favor of the involvement of alternative non-canonical autophagy, distinct from LAP, in the elimination of the parasite during this late hepatic stage [37,41]. Interestingly, prolonged expression of LC3-II molecules has been observed on the PVM [41]. The lipidation and incorporation of LC3-II at the PVM is promoted by Atg5, and is necessary for the subsequent binding of ubiquitin, sequestosome 1 (Sqstm11/p62), and neighbor of *BRCA1* gene 1 (Nbr1) at the membrane [42]. Boonhok et al. (2016) have described an interferon-gamma (IFN-γ)-mediated non-canonical autophagy machinery for the elimination of *P. vivax* sporozoites in primary human hepatocyte cultures, called *Plasmodium*-associated autophagy-related (PAAR) responses (Figure 1A) [43]. PAAR is characterized by the expression of MAPLC3/LC3 on the PVM, and is dependent upon Becn1, PI3K, and Atg5, but not Ulk1 [43]. Although some intra-hepatocyte parasites are eliminated by LAP-like autophagy, others can escape cellular autodigestive elimination [41]. Indeed, to control and avoid elimination by autophagy, mature schizonts remodel LC3^+^PVM by shedding autophagic proteins trapped in the tubovesicular network (TVN), and by redirecting the lysosome to the TVN [36,39]. In addition, the interaction of *PbA*, upregulated in infective sporozoites gene 3 (UIS3) protein, with host MAP1LC3/LC3 inhibits the autophagy function, and allows parasite replication [38,44]. In parallel, *Pb*A is able to divert the non-selective host autophagy and exploits it as a nutritive source for supporting parasite growth [41,45,46]. Together, these mechanisms favor the egress of hepatic merozoites, which are released from infected hepatocytes as merosomes, another pMV that consists of hundreds of parasites surrounded by a host cell membrane [47,48,49,50]. During this process, *Plasmodium* seems to use vesicle encapsulation as a mechanism to hide from host immune mechanisms, and to escape elimination. 

### 3.3. Autophagy and pMV Crosstalk during Blood-Stage Infection

Merozoites eventually released from merosomes invade RBCs to initiate the blood stage of infection [3,51]. The parasite uses its autophagy machinery (*Pf*ATG) to recycle unnecessary proteins to support its nourishment and biogenesis (Figure 1B). *Pf*ATG also allows for survival in the case of nutrient starvation, especially in the intraerythrocytic stage [52]. However, unlike hepatocytes, RBCs do not generate intracellular autophagic host defense mechanisms for parasite clearance [52,53,54]. Interestingly, merosomes are potent immunoregulators. They induce macrophage activation via Toll-like receptor 4 (TLR4) and myeloid differentiation primary response 88 (MyD88) pathways, resulting in an increase in CD40 expression and tumor necrosis factor alpha (TNF-α) secretion [55]. Blood-stage *P. falciparum*-derived MVs (*Pf*MVs) express parasite antigens capable of stimulating human peripheral blood mononuclear cells (PBMCs) and macrophages, upregulating pro-inflammatory cytokine secretion and activating neutrophil migration [20]. In *P. vivax*-infected patients, circulating pMVs can adhere to human splenic cells by interacting with intercellular adhesion molecule-1 (ICAM-1) [56]. Recently, *Pf*MVs have been shown to induce natural killer (NK) cell responses during malaria, after interacting with the melanoma differentiation-associated protein 5 (MDA5), a RIG-I-like receptor (RLR) [57]. PMVs isolated from *P. falciparum*- and *P. vivax*-infected patients have been shown to transport miRNA and parasite proteins [14,58]. Such vesicles can be internalized by iRBCs, resulting in the transfer of genetic information inside exosome-like vesicles, and these vesicles have been found to promote gametocytogenesis [20,59]. In addition, pMVs derived from human brain endothelial cells stimulate the proliferation/activation of T cells [60,61]. These later observations reinforce the hypothesis of a role for pMVs in the pathogenesis of CM.

### 3.4. Role of Autophagy-Dependent pMVs in CM

CM is a lethal complication of *Plasmodium* infection. It causes around 0.4 million deaths per year, principally in children <5 years of age and in immunosuppressed individuals [1]. The pathophysiology is a consequence of the sequestration of iRBCs in brain microvessels and the detrimental immune response, characterized by neuroinflammation and the lymphocyte migration that results from it [62,63,64,65]. In C57BL/6 CM-susceptible (CM^S^) mice infected with *Pb*A, astrocytes are activated early during infection, and release pro-inflammatory factors that contribute to neuroinflammation and death [66,67,68]. We have previously described the in vitro transfer of *Pb*A-MVs to astrocytes after contact with iRBCs (Figure 1C) [68]. We propose that iRBCs located in the perivascular space could interact with pseudopodia from activated astrocytes during neuropathogenesis [66,69]. As shown in Figure 2A, the transfer of *Pb*A-MVs occurs at a contact point between the astrocytic foot and *Pb*A-iRBCs in CM^S_^derived primary astrocyte cultures stimulated for 6 h. This transfer of *Pb*A-MVs to astrocytes occurs via the unconventional LAP autophagy pathway, which is independent of Ulk1 and calcium binding and coiled-coil domain 2 (Calcoco2/Ndp52) [28] (Figure 1C). The LAP pathway is activated via Becn1, the RUN domain, and the cysteine-rich domain, containing Beclin 1-interacting protein (Rubcn), Atg16L1, Atg5, and Sqstm1/p62. The parasite material transferred to the astrocyte is contained in PV-expressing LC3-II molecules after activation. LAP mediates the degradation of *Pb*A-MVs in astrocytes (Figure 2B,C). Eventually, an organelle called LAPosome fuses with lysosomes to degrade *Pb*A-MVs (Figure 2B,C). The treatment of *Pb*A CM^S^ mice with bafilomycin A_1_, an autophagy inhibitor, has been shown to prevent ECM [28]. Confocal microscopy revealed that, in contrast, in CM^S^-derived astrocytes in CM^R^-derived cells, *Pb*A-MVs transferred from iRBCs were not internalized, but, instead, remained at the astrocyte extracellular membrane (Figure 2B). Indeed, RT-qPCR only detected a small quantity of *P. berghei 18S ribosomal* (*PB18S*) gene in these astrocytes (Figure 2C). Unlike in the case of CM^S^-derived astrocytes, increased expression of *MAP1LC3/LC3*, *ATG5* and *ATG16L1* LAP autophagy genes was not observed in CM^R^-derived cells (Figure 2D). 

Astrocytes are key players in the innate immune response in the brain [68,70,71]. During malaria, they secrete pro-inflammatory cytokines/chemokines, and express major histocompatibility complex (MHC) class I and II molecules [66,67,68]. We demonstrated that the secretion of chemokine (C-X-C motif) ligand 10 (CXCL10), chemokine (C-C motif) ligand 2 (CCL2), and TNF-α by astrocytes is dependent on the transfer and degradation of *Pb*A-MVs by the LAP pathway [28]. It is known that the production of CXCL10 in the brain is a prerequisite for the recruitment of effector CD8^+^ T cells expressing the chemokine (C-X-C motif) receptor 3 (CXCR3) involved in ECM [65,67,72,73]. We also evidenced that astrocytes from CM^R^ mice produced significantly lower levels of pro-inflammatory cytokines/chemokines than CM^S^-derived cells after stimulation with *Pb*A-iRBCs (Figure 3A). This is probably due to the smaller number of *Pb*A-MVs transferred to astrocytes from CM^R^ mice. However, we also observed upregulation of *leukaemia inhibitory factor* (*LIF*), *transforming growth factor beta* (*TGF-β*), and *interleukin 10* (*IL10*) gene expressions in CM^S^-derived astrocytes after contact with *Pb*A-iRBCs (Figure 3A). 

Activated astrocytes express pattern recognition receptors (PRRs), such as TLRs, which are key sensors of danger, and are involved in the initiation of the brain’s innate immune response [74,75,76]. After 24 h of parasite contact, we observed upregulation of *TLR3* gene expression in CM^S^-derived astrocytes (Figure 3B). The expression of genes of pro-inflammatory signaling molecules, such as *TIR domain-containing adapter-inducible interferon-β (TRIF*), *TNF receptor-associated factor 6* (*TRAF6*), *TRAF3, TANK-binding kinase 1* (*TBK1*), *interferon regulatory factor 3* (*IRF3*), *IRF7*, and *IFN-β*, was also increased (Figure 3B). In contrast to what was observed in astrocytes derived from CM^S^ mice, in ECM-resistant TLR3 knockout (KO) mice, the expression of *PB18S* and autophagy genes was significantly decreased following *Pb*A-MVs transfer in astrocytes (Figure 3C,D). In addition, the production of CXCL10, CCL2, TNF-α, IL-10, and TGF-β was totally abolished in astrocytes from TLR3KO mice, unlike in cells from CM^S^ mice (Figure 3E). These observations strongly suggest a link between the LAP pathway and TLR3-dependent pathway in the transfer of *Pb*A-MVs and the induction of pro-inflammatory cytokine and chemokine responses in astrocytes from CM^S^ mice. Therefore, a TLR3-TRIF-dependent pathway could also participate in neuroinflammation during ECM, similar to what has been reported for the intrahepatic stage [77]. 

## 4. Discussion

This review examines the role of autophagy in the different developmental stages of the *Plasmodium* parasite. Autophagy is involved in a multitude of parasite biological processes, such as the regulation of intracellular cytoplasmic protein turnover, organelle differentiation, parasite growth, gametogenesis, and infection dissemination. By virtue of its involvement in the biogenesis and function of *Plasmodium*-derived EVs, autophagy also influences the stimulation of pro-inflammatory responses of innate immune cells and, hence, the severity of the disease. 

Our recent description of the involvement of LAP in the transfer of *Pb*A-derived MVs to astrocytes, and the resulting induction of pro-inflammatory factors, exemplifies the importance of MVs in host–parasite interactions and infection outcomes. We also found that a TLR3-mediated anti-inflammatory response is induced in astrocytes after contact with parasite MVs. These observations suggest a dual, but contrasting, role for autophagy in parasite–astrocyte interactions. Autophagy can favor either (i) detrimental neuroinflammation, through the production of CXCL-10 that exacerbates the inflammatory process, or (ii) a protective outcome, resulting from the production of neuroimmunomodulators, such as LIF, which downregulate the exacerbated inflammation and prevent T cells from infiltrating the brain [78]. Through the genesis of parasite-derived EVs, LAP may also influence antigen presentation by astrocytes to pathogenic CD8^+^ T cells that have migrated to the brain. It is important to note that the astrocytes’ production of pro-/anti-inflammatory factors is regulated by crucial molecular events that are responsible for neuropathology during malaria. These events need to be explored further (Figure 4). 

In summary, *Plasmodium*-derived MVs interact with autophagy pathways to contribute to protection/pathology during malaria. On the one hand, they promote parasite clearance, stimulate a pro-inflammatory innate immune response, and contribute to the downregulation of brain inflammation during CM. On the other hand, they participate in the dissemination of the parasite within the host, as well as in its differentiation to sexual forms. They also contribute to the activation of pro-inflammatory innate responses. Besides this, they can precipitate severe disease by promoting antigen presentation to pathological CD8^+^ T cells by astrocytes that infiltrate the brain during CM.

## Figures and Tables

**Figure 1 life-12-00415-f001:**
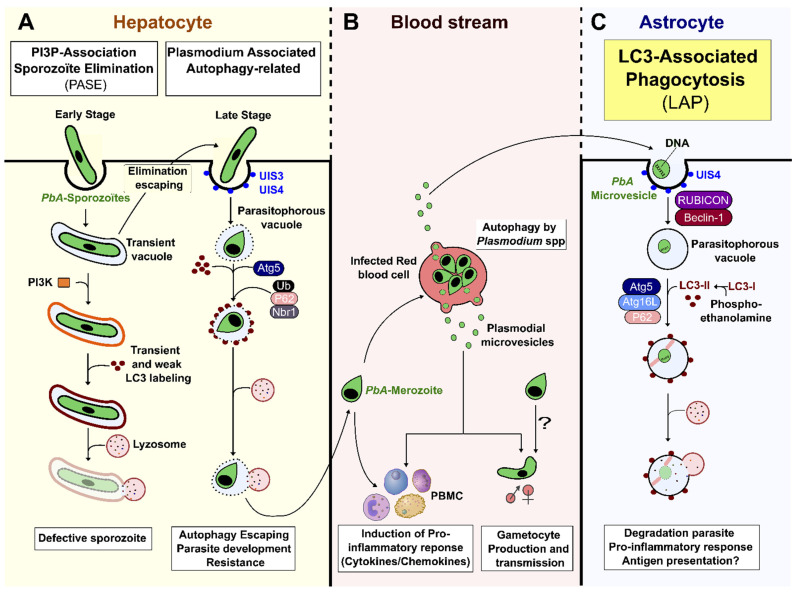
Host autophagy pathways and *Plasmodium*-derived microvesicles. (**A**) *Plasmodium* sporozoite infects a hepatocyte by invagination of the host cell membrane, thus forming a transient vacuole (TV) or parasitophorous vacuole (PV). If parasites are unable to discard their unnecessary organelles, or correctly remodel their vacuole, the PI3K complex is formed at the vacuole membrane, and leads to parasite elimination by cytosolic lysosomes. Although LC3 is observed transiently at the vacuole membrane, it is not essential for efficient *Plasmodium* degradation. This elimination, called PI3P-associated sporozoite elimination (PASE), occurs during the early intrahepatic stage. During the late intrahepatic stage, when the parasite starts its differentiation and multiplication, the PVM enclosing the parasite can be labelled by LC3 promoted by Atg5, resulting in the host *Plasmodium*-associated autophagy-related (PAAR) response. This pathway, associated with ubiquitin, sqstm1/p62, and Nbr1, is independent of PI3k, Rb1cc1, and Ulk complexes, and does not necessarily lead to clearance of the parasite, as it can avoid fusion with lysosomes by remodeling its vacuolar membrane. (**B**) After leaving the liver, merozoites invade RBCs. The parasite is able to produce *Plasmodium*-derived microvesicles by autophagy, using *PbA*ATG, in order to transport genetic information and promote gametocytogenesis. However, pMVs can also induce a pro-inflammatory response. (**C**) *Pb*A-MVs are transferred from iRBCs to astrocytes inside a PVM directly targeted by LC3-II to form a LAPosome. This then fuses with lysosomes, resulting in parasite clearance. This LC3-associated phagocytosis (LAP) pathway, an unconventional autophagy pathway, induces a pro-inflammatory response in astrocytes, leading to ECM.

**Figure 2 life-12-00415-f002:**
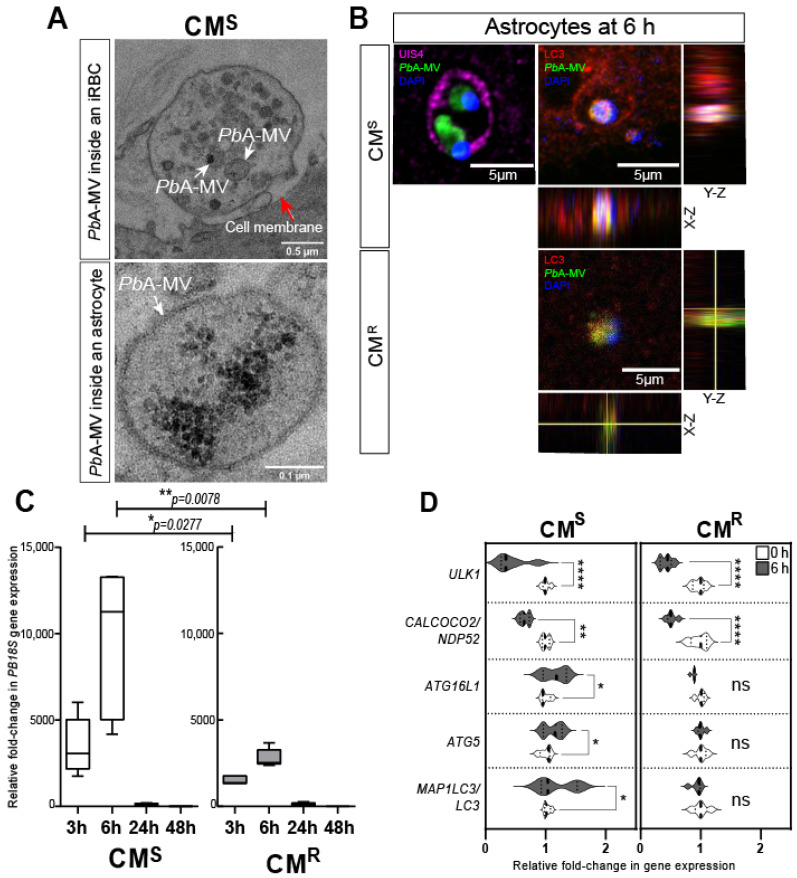
*Pb*A-MVs transfer to astrocytes from ECM-sensitive (CM^S^) and -resistant (CM^R^) mice upon 6-h contact with iRBCs. Primary astrocyte cultures derived from neonatal CM^S^ or CM^R^ mice were stimulated for 6 h with GFP-*Pb*A-iRBCs, washed, and followed for 48 h post iRBC contact, as previously described [28]. (**A**) Transmission electron microscopy revealed that *Pb*A-MVs were transferred to CM^S^-derived astrocytes at the contact point with iRBCs (top micrograph, white arrows) and were observed intracellularly (bottom micrograph, white arrows) at the 6-h time point. The red arrow shows the cell membrane of the astrocyte. (**B**) GFP-*Pb*A-MV (green) and parasite DNA (blue) are enclosed inside a PV labelled by UIS4 (pink; left panel) or LC3-II (red; right panel) to form a LAPosome inside CM^S^-derived astrocytes. By contrast, *Pb*A-MVs remained at the cell membrane of CM^R^-derived astrocytes. (**C**) The quantity of *PB18S* gene detected in CM^R^-derived astrocytes was significantly lower than that found in CM^S^-derived cells, confirming reduced transfer of *Pb*A-MVs in these cells. (**D**) LAP-related gene expression did not increase in CM^R^-derived astrocytes, as compared to CM^S^-derived cells, 6 h after *Pb*A-iRBC contact. Student’s *t*-test was used to compare median fold change in gene expression in panels C and D (*n* = 5). * *p* < 0.05; ** *p* < 0.01; **** *p* < 0.0001.

**Figure 3 life-12-00415-f003:**
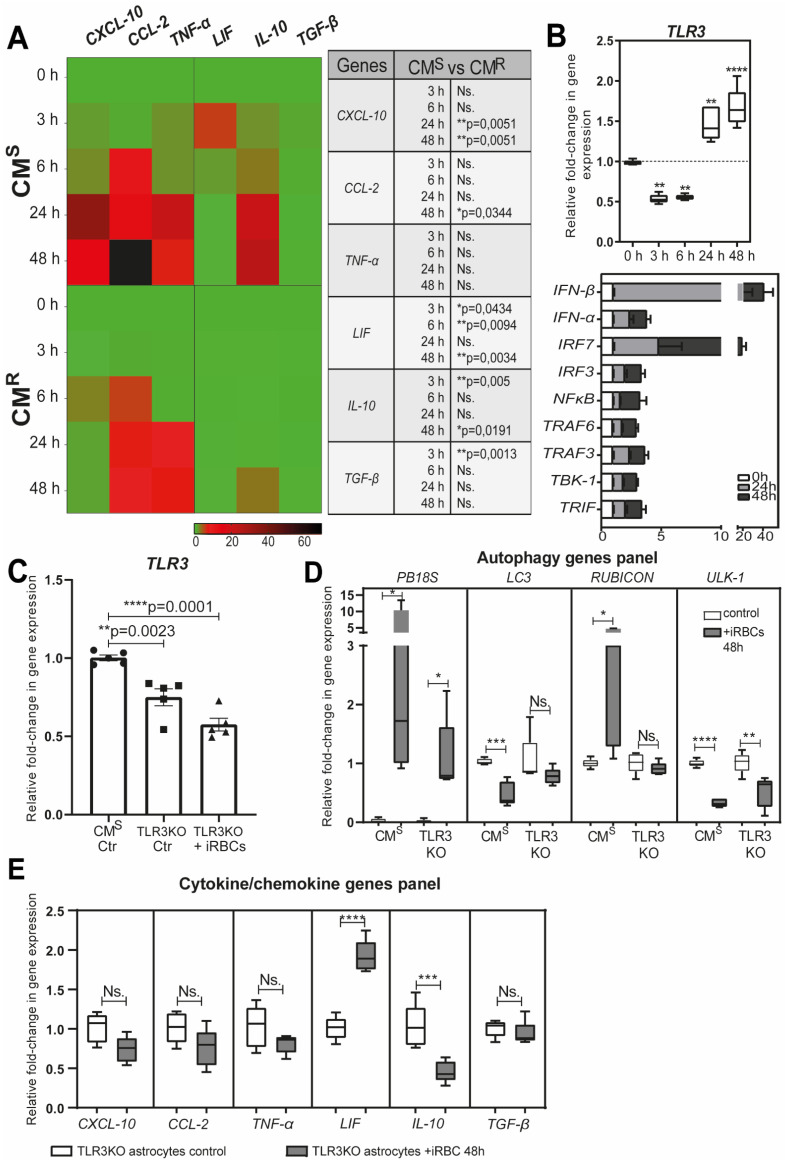
Immune response of astrocytes after *Pb*A-MV transfer and TLR3 engagement. Primary astrocyte cultures, derived from CM^S^, CM^R^, or TLR3KO neonatal mice, were stimulated for 6 h with GFP-*Pb*A-iRBCs. The cells were then washed and followed up for 48 h post iRBC contact. (**A**) Pro-inflammatory *CXCL-10*, *CCL-2,* and *TNF-α* genes and anti-inflammatory cytokine *LIF*, *IL10,* and *TGF-β* genes were highly expressed in CM^S^-derived astrocytes, as compared to CM^R^-derived cells. (**B**) TLR3 pathway genes were significantly upregulated in CM^S^-derived astrocytes after 24 h of contact with *Pb*A-iRBCs. (**C**) *TLR3* gene expression was totally abolished in TLR3KO-derived astrocytes. (**D**) Decreased *PB18S* gene expression and downregulation of autophagy-related genes (*LC3*, *RUBCN*, and *ULK1*) were observed in TLR3KO-derived, but not CM^S^-derived, astrocytes at 48 h after iRBC stimulation. White bars indicate 0-h stimulation and grey bars indicate 48-h stimulation. (**E**) The absence of *CXCL-10*, *CCL-2*, *TNF-α*, *IL-10*, and *TGF-β* gene expression in TLR3KO-derived astrocytes suggests involvement of the TLR3 pathway in the astrocyte immune response. Student’s *t*-test (except for (**B**), where one-way ANOVA was used) was used to compare median fold change (*n* = 5). * *p* < 0.05; ** *p* < 0.01; *** *p* < 0.001; **** *p* < 0.0001.

**Figure 4 life-12-00415-f004:**
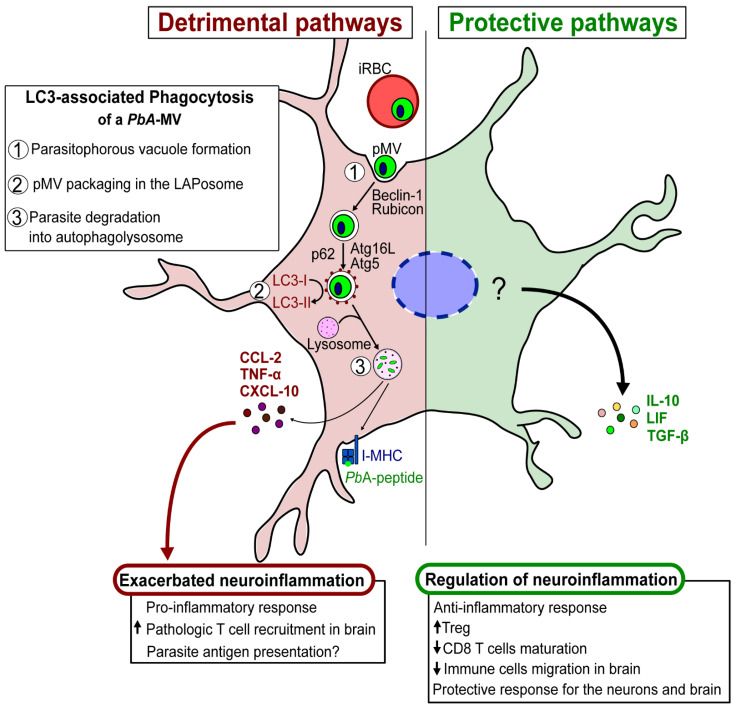
Schematic hypothesis of detrimental/protective pathways of astrocytes involved in the pathogenesis of ECM.

## Data Availability

Not applicable.

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
