# Peer review of "Autophagy Pathways in the Genesis of Plasmodium-Derived Microvesicles: A Double-Edged Sword?"

_life, 2022, doi:10.3390/life12030415_

Round 1

Reviewer 1 Report

It is a nice review on the dual but contrasting role for autophagy in parasite–astrocyte interactions. It can be published without major changes.

Minor suggestions:

Use italic letters for genus and species names also in the reference list.

Ref 32: Mota M.M et al

line 24: than instead of that

Some abbreviations are unnecessary. E.g., PASE occurs only twice, one occurrence in the text, other one in the legend for Fig. 1, and in both cases the full name is also given. As a “fist rule”, do not abbreviate words or expressions except they occur at least three times. There are abbreviations in the paper which occur only once or twice (e.g., TVN).

Author Response

We thank the reviewers for their critical evaluation and supportive comments and suggestions. You will find a point by point answer to reviewers' comments:

Reviewer 2 Report

  1. Several studies reported association of LAP autophagy with reduced neuroinflammation. How do the authors address this conflicting findings in case of PbA-MVs?
  2. In fig 2A in the panel for CMs the 2 indicated PbA-MVs look very different from each other. Is that a typographical error or if they are actually so different in form and size what could be the possible reason for this?
  3. High resolution images for figure 2B would be appreciated.
  4. Do the data shown in the article belong to unpublished studies by the authors or are they cited from the published articles? If it's unpublished where is the methodology section and if it's from published articles the references haven't been cited properly. It is confusing as a reader since the authors mention it as a review.
  5. Grammatical proofread is required.

Author Response

(The authors gave the same response as above.)

Round 2

Reviewer 2 Report

Thank you for clarifying all the queries and confusions about the article and incorporating the necessary changes. This improves the overall value and comprehensiveness of the manuscript for the readers.